# Using ELEFIGHT^®^ QR Codes for Quick Access to Information on Influenza Burden and Prevention: A Pilot Study in Lyon University Hospital

**DOI:** 10.3390/vaccines10101591

**Published:** 2022-09-22

**Authors:** Nagham Khanafer, Sylvain Oudot, Catherine Planckaert, Nathalie Paquin, Camille Mena, Nadège Trehet Mandel, Roland Chapurlat, Catherine Lombard, Géraldine Martin-Gaujard, Laurent Juillard, Christelle Elias, Audrey Janoly-Dumenil, Anne Jolivot, Meriem Benazzouz, Margot Maligeay, Marie-Pierre Ayala, Diana Ismail, Philippe Vanhems

**Affiliations:** 1Unité D’hygiène, Epidémiologie et Prévention, Hôpital Edouard Herriot, Hospices Civils de Lyon (HCL), 69003 Lyon, France; 2Equipe PHE3ID, Centre International de Recherche en Infectiologie (CIRI), Inserm U1111, CNRS UMR5308, ENS de Lyon, Université de Lyon1, 69007 Lyon, France; 3Sanofi, 69437 Lyon, France; 4Hôpital Edouard Herriot, Hospices Civils de Lyon (HCL), 69437 Lyon, France; 5Inserm, F-CRIN, Réseau Innovative Clinical Research in Vaccinology (I-REIVAC), 75679 Paris, France

**Keywords:** influenza, flu, VCR, QR code, hospital, prevention, nudge, vaccination, reminder, ELEFIGHT

## Abstract

(1) Background: The Vaccine Coverage Rate of influenza remains low and omnichannel efforts are required to improve it. The objective was to evaluate the feasibility and outcomes of a QR Code nudging system in outpatient departments. (2) Methods: The study was performed in 6 departments ensuring ambulatory activities in a French university Hospital between November and December 2021. By scanning QR codes, users accessed anonymously to the ELEFIGHT^®^ web app, which provides medical information on influenza and invites them to initiate a discussion about influenza prevention with their physicians during the consultation. (3) Results: 351 people made 529 scans with an average reading time of 1 min and 4 s and a conversion rate of 32%, i.e., people willing to engage in a discussion. (4) Conclusions: The study suggests that direct access to medical information through QR codes in hospitals might help nudge people to raise their awareness and trigger their action on influenza prevention.

## 1. Introduction

Worldwide, annual influenza epidemics result in 1 billion influenza infections with 290,000 to 650,000 deaths, mainly among older adults and people at risk [1]. Influenza vaccines are effective in decreasing hospitalizations and mortality due to influenza and its complications [2]. The French Agency of Public Health (Santé Publique France) promotes influenza vaccination amongst the public at large as well as health care workers (HCWs) with a goal to increase the vaccine coverage rate of influenza (Flu VCR) in people over 65 and those with a high risk of influenza complications [3]. Currently, interventions are mainly based on the national health insurance system that directly supports vaccination of these specific groups. In addition to campaigns on TV, the French health insurance sends vaccine invitations or vouchers to the target population. However, the Flu VCR in France is far below the 75% objective set by the World Health Organization [4,5].

This low rate highlights that relying on the traditional primary care setting for adult vaccination may not be sufficient to achieve the national goal of VCR. One option to increase uptake among the risk groups, as well as the HCWs, might be to promote vaccination by providing digital motivational messages and creating opportunities for sharing preventive messages in various health care settings contexts, such as hospitals [6].

A certain number of interventions for prevention have been tested, mainly those supported by using smartphones, patient monitoring devices, and other wireless sensors [7].

In recent years, some institutions have adopted the use of two-dimensional bar codes, also known as QR (Quick Response) codes, for encoding information available through smartphones, tablets, and other electronic devices. These QR codes can serve as a vehicle to accelerate the real-time accessibility of information and facilitate individual decision-making in many fields, including health [7]. QR codes can be easily generated without any costs via an application or a website [8] and displayed on posters or screens.

To the best of our knowledge, no data are available on the potential benefit of this technology regarding influenza burden awareness and its prevention by vaccination at hospitals. On top of conversations with HCWs, it is highly probable that patients might gain from additional information accessible to them through scanning the QR code during any free time. Direct access to validated information on this infection and its vaccination would optimize the discussion between patients and their physicians and hospital pharmacists.

The objective of this pilot study was to explore the benefits of using QR codes to enhance awareness of hospital outpatients on influenza disease and vaccination and to identify perceptions towards their use during implementation, leveraging ELEFIGHT^®^ anonymous digital solution.

## 2. Materials and Methods

ELEFIGHT^®^ is a web application that aims at sharing validated medical information about a disease or a risk to the lay public or HCWs, to encourage behavioral changes through the concept of nudge. A “call to action” button is floating over the text to suggest the action to be made (e.g., ask for more information, watch a video, talk to the HCWs, etc.). By providing validated information at a place where it matters (e.g., at a hospital or in a pharmacy), the users are suggested to become actors in their own health. They may take action leading to better prevention that directly benefits them and the community. As a web app, no download of ELEFIGHT^®^ is needed. ELEFIGHT^®^ is a web app, i.e., the visitor connects to a website with an app-like page. Upon connecting, a cookie policy is displayed: “We use non-sensitive information like cookies or device identifiers for purposes like measuring traffic and preferences of our visitors as well as personalized content.” The visitor has the choice to accept the cookie or not, but in either case, he has access to the same content. The visitor is always ANONYMOUS.

For this reason, no personal information is requested. ELEFIGHT^®^ pilot study was carried out at Edouard Herriot Hospital, an 850-bed University Hospital in Lyon, Hospices Civils de Lyon, Lyon, France, between November 8 and 10 December 2021, in the departments ensuring ambulatory activities for adult patients, i.e., general internal medicine, oncology, nephrology, rheumatology, geriatrics, and pharmacy. Separate QR codes were printed on the posters installed in the waiting rooms of these departments. A specific QR code was also developed for HCWs reserved spaces, with the same content as for patients. Once a week, a clinical research associate (CRA) went to the waiting rooms (except in pharmacy) to explain the ELEFIGHT^®^ pilot study to patients and encourage them to scan the QR code with their smartphones.

The study was planned to start at the same time as the official French National flu campaign. The launch day was kept unchanged for our study, but the National campaign had been unexpectedly moved forward.

After scanning the QR codes, the ELEFIGHT^®^ web app invites the users to initiate a discussion about influenza vaccination with their respective physicians during the consultation—by offering them a list of potential questions they would like to ask after clicking on the call-to-action button “Talk to my doctor.”

The following data were computerized anonymously in real-time:(1)Number of users who scanned the QR code;(2)Bouncing rate (users who closed the web app right after the opening);(3)Average connection time to the web app;(4)Number of clicks on the “ Talk to my doctor” button, and(5)Conversion rate: number of users who clicked the “Talk to my doctor” button divided by the number of users who scanned the QR code and stayed on the web app.

## 3. Results

Between November 8 and December 10, 2021, 351 people (219 patients and 132 HCWs) realized 529 scans of QR codes with an average conversion rate of 32%. The number of scans across time is described in Figure 1.

Most scans were done by patients consulting in general internal medicine (39%) and by HCWs (38.5%). For the other departments, the rates of participation were 6.3% in nephrology, 5.6% in oncology, 4.6% in pharmacy, 3.3% in geriatrics, and 2.7% in rheumatology.

People read the information about influenza disease and vaccination on an average of 1 min and 4 s. This is coherent with the amount of content that was made available to them. Figure 2 describes the number of scans and of those who clicked to read the information on influenza disease and prevention.

The proportion of users who clicked the “Talk to my doctor” button was higher in patients compared with HCWs. These data in both populations are described in Figure 3A,B.

Moreover, the added value of the CRA to motivate patients to scan the QR code is depicted in Figure 4. The presence of the CRA was associated with an increase in performed scans. That suggests the need to cooperate with patients to initiate proactive action.

## 4. Discussion

The purpose of this project was to deploy a pilot study that could help patients to be informed about the risks related to influenza and to be suggested to engage in discussion with their hospital’s treating physicians about the disease and its prevention through vaccination. The experiment was well accepted by the patients, and it shows that the number of scans was higher when the CRA was present. The access to information regarding influenza and its prevention by scanning QR codes delivering medical information based on French recommendations and the nudge with the call-to-action button was tested in this ELEFIGHT^®^ pilot study. The benefits of the annual influenza vaccine are well documented at the individual and public health levels. However, multipronged efforts remain needed to achieve the expected Flu VCR. In a recent review, some hospital-based interventions seem effective in improving the inpatient vaccination [6]. Individual-level process-driven strategies (i.e., order protocols, reminders) dominate this literature, but the few studies which examined hospital reward programs, interdepartmental competitions, hospital-wide campaigns, and staff education showed success [6]. As the population ages and cost-pressure increases, the health care system continues to face challenges and must find ways to enhance the role of patients in the management of their diseases [9].

The French National influenza campaign had been launched two weeks in advance of the announced initial date. Consequently, the ELEFIGHT^®^ pilot study started two weeks after the start of the National influenza campaign, as the start date could not be moved earlier.

The study was conducted in departments ensuring ambulatory activities for adult patients. No exclusion was intentionally made, but in a few departments (e.g., ophthalmology), the study was not particularly convenient for their daily practice. Moreover, the hospital pharmacy delivered medications for patients mainly suffering from chronic diseases for whom vaccination is highly recommended.

This experiment showed that some participating patients did not scan the QR code without the intervention of CRA. This can be related to different reasons. First, patients did not see the poster; the sober design (white and pastel colors) may make the poster less visual than the other information posters. Second, it could be that they did not know a QR code could be scanned and provides additional content. Finally, it could be associated with a lack of interest in the influenza topic in November/December 2021, during the new COVID19 wave in France, or a few weeks after the start of the National Flu vaccination campaign.

The role of the CRA was to explain the project, draw the patient’s attention to the posters and show how to scan the QR code with a smartphone. Nevertheless, we can note that the posters were left after the experiment, i.e., 60 scans were computerized between 10 and 18 March 2022 without the presence of a CRA. At that time, the increase in influenza cases was discussed in the French Media. It suggests that this overwhelming communication on the influenza disease may have generated interest in this topic, indirectly leading people to get more information by scanning a QR code on a poster. Another ELEFIGHT^®^ pilot study conducted in a similar context in the United Arab Emirates in January 2021 without any intervention of a CRA, showed very close numbers of scans and conversion rates (unpublished data).

This suggests that the design of the poster (e.g., colors, size), the location, and the communication could be levers to increase its adoption.

Some patients declared a low interest in influenza disease and influenza vaccination but would be ready to test the digital tool for another medical subject. By identifying underlying reasons for less interest in influenza vaccination, social elements that need targeting may be identified and could guide future interventions or policy development to achieve vaccination goals [10].

People who were already vaccinated against flu were less inclined to scan the QR codes to obtain additional information on influenza. Another observation concerns older adults who are rarely equipped with a smartphone at all or one suitable for reading QR codes. In addition, we confirmed that perception of QR code technology and its usage could be affected by certain demographic factors such as age [11]. Therefore, efforts might be made to facilitate access to the web app, regardless of age or other patient determinants, e.g., tablets available for public access in the waiting rooms.

The conversion rate of 32% in this ELEFIGHT^®^ pilot study supports that nudging through QR codes for increasing patient awareness about influenza and Flu vaccination is feasible in hospital settings. This rate may be partly accounted for by (1) demographic differences in sentiments about influenza risks across participants due to the absence of influenza epidemics over the last two years, (2) differences in the patients’ perception of the risks associated with influenza, (3) emotional and social fatigue about vaccination and risks related to viral infectious disease caused by COVID-19 pandemic discussions, and (4) technical problems (i.e., the availability of 3G/4G/5G internet network, lack of awareness on how to scan with a smartphone). In fact, as with many new technologies, QR codes have some usability issues linked to access to the internet or dexterity in the usage of a smartphone. For all their opportunities and strengths, they have a straightforward weakness: requiring a smartphone or a tablet with a camera [8].

The provision of reliable patient education is essential for shared decision-making. The consultation/reading of educational materials, accessed during the waiting time prior to consultation or prescription medicine supply (hospital pharmacy), can potentially benefit clinical practice if implemented in a time- and resource-efficient manner [12].

## 5. Conclusions

In conclusion, this experiment suggests that a facilitated direct access to medical information through QR codes disseminated in health settings can help nudge people to raise their awareness of influenza and its prevention and act through a call-to-action option. The ELEFIGHT^®^ pilot study suggests that patients affected or exposed to other diseases might also benefit from the use of the ELEFIGHT^®^ digital solution.

## Figures and Tables

**Figure 1 vaccines-10-01591-f001:**
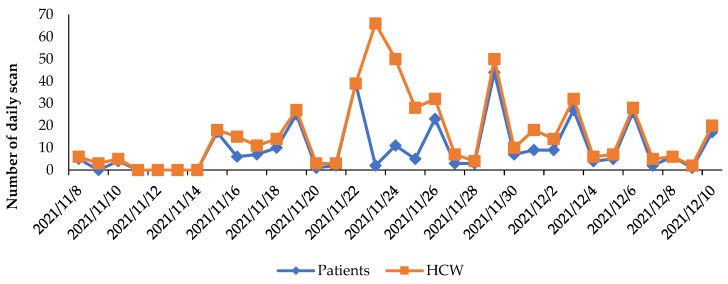
Daily number of scans realized by patients and HCWs during the study period.

**Figure 2 vaccines-10-01591-f002:**
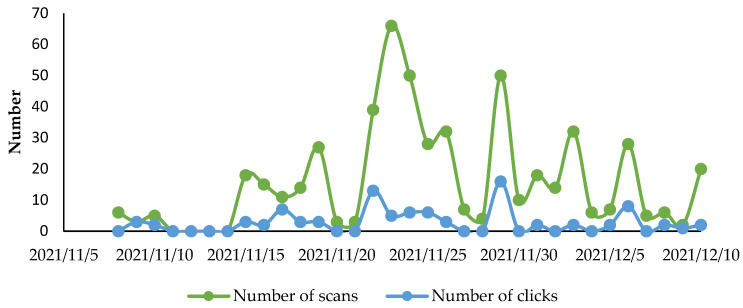
Numbers of scans and clicks realized by patients and HCWs.

**Figure 3 vaccines-10-01591-f003:**
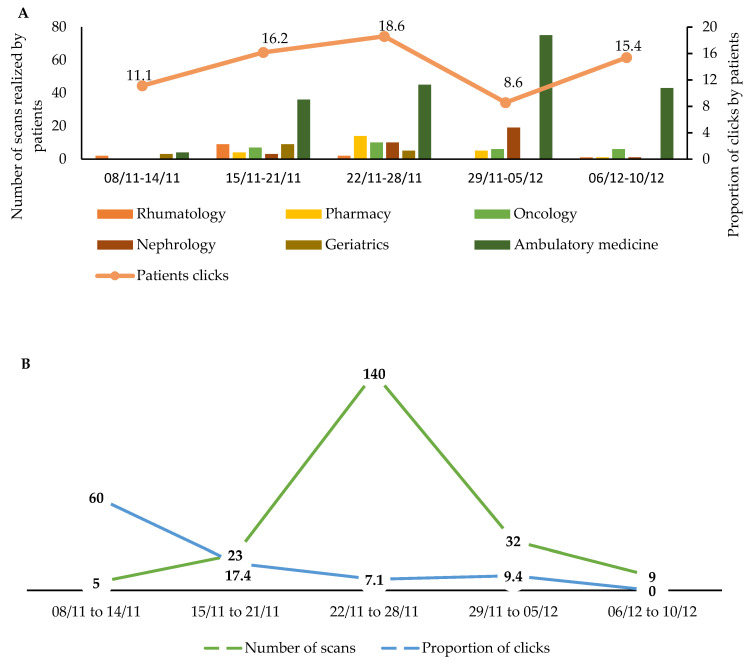
Numbers of scans and proportion of clicks realized by patients (**A**) and HCWs (**B**).

**Figure 4 vaccines-10-01591-f004:**
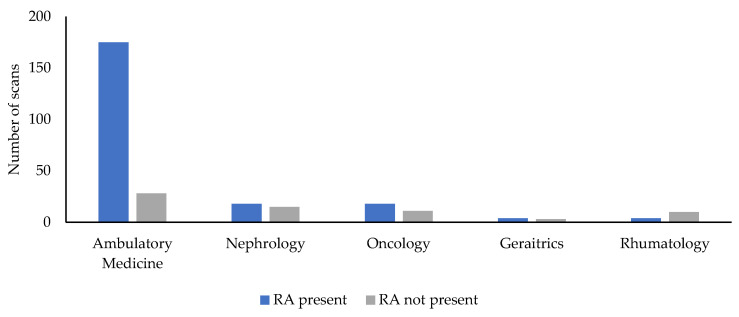
Numbers of scans in outpatients by clinical wards and the presence or not of the clinical research assistant.

## Data Availability

Not applicable.

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
