# Peer review of "Using ELEFIGHT® QR Codes for Quick Access to Information on Influenza Burden and Prevention: A Pilot Study in Lyon University Hospital"

_vaccines, 2022, doi:10.3390/vaccines10101591_

Round 1

Reviewer 1 Report

The article is well written and concise. The information is clearly understood, is presented straightforward and summarized succinctly. There are only a few items to address, otherwise the authors have submitted a nicely composed article.

Page 2, line 50. ‘Campaign’ should be ‘campaigns’

Page 2, lines 86-87. Either explain how it guaranteed the privacy of users and/or provide a reference please.

Page 2, lines 97-98. Why did the campaign launch 2 weeks early, it isn’t addressed in the text.

All Figures. In order to better define the difference between the lines and bars of the various charts, colour might be useful to incorporate. As it stands, it’s easy to get lost in some areas due to the similarity between the grey and black. Colour would clarify that issue.

Discussion. Would future plans for public information display include placement of posters at public venues such as metro and train stations? What are the future ideas are you considering to engage with the public?

Conclusion. Are there any potential plans to expand this website to include other types of health information? Or possibly collaborate with other countries that would benefit from better health information distribution?

Author Response

Please find attached our responses to the reviewers and a revised version of our manuscript entitled “Using ELEFIGHT® QR codes for a quick access to information on Influenza burden and prevention:  a pilot study in Lyon University Hospital”.

We were pleased to have received your pertinent comments. The modifications are in red in the revised version submitted today. Any additional requests regarding the revised version will be welcome.

All of the authors have approved the new submission of the manuscript and we assure you that it is not considered for publication elsewhere.

Reviewer 2 Report

The submitted manuscript by Khanafer et al. “Using ELEFIGHT® QR codes for a quick access to information on Influenza burden and prevention: a pilot study in Lyon University Hospital” aimed at exploring the benefits of using QR codes to enhance awareness of hospital outpatients on influenza disease and vaccination and to identify perceptions towards their use during implementation.

I feel that the manuscript in its current version is not suitable for publication in the Vaccines journal, and that it would benefit of a major revision. Below, I point out to some major issues and provide point-by-point comments.

Primarily, I encourage Author to consider changing type of the paper from “Article” to “Short Communication” or similar, if more appropriate, since these are the results from a pilot study, and on quite modest sample with the modest results.

Paper is well written, results are fairly modest but nicely presented, but my main concern is that the aim of this study is not completely fulfilled, i.e. there was no evaluated of actual effect of using this approach to access information (for example, pre- vs. post- app using vaccination uptake; evaluating pre- vs. post- intervention knowledge and attitudes on fly prevention, vaccination; or similar) where it could be explored if something actually changed (i.e. whether there was an effect) after scanning QR codes /clicking. Here is presented how many participants access information during (short) study period.

Being the pilot study, if there is an actual study conducted afterwards, these data should be presented together (hopefully in more detail and with some additional statistical analyses) in order to increase the quality of the paper.

Since the data was anonymous, we unfortunately cant know the demographic of the participants in particular is important the age structure of participants which might be quite relevant here when drawing conclusion since elderly (which are very important target group for the influenza vaccine) might not be enough familiar or equipped with technology in order to participate and benefit from this approach.

Was there any control whether the same participant entered several time on the web-app (i.e. recognizing the IP address or similar) during the study period? if yes, this should be described.

it seems unclear why there were two separate codes (for patients and for HCW) and how was this controlled? I mean, patients could have clicked/scanned both, just for curiosity, or not reading detailed info, or similar? it has space for misclassification here? this should be better explained?

Also, it remains unclear which hypotheses were tested within this study? if any? no statistical methods were used? Another argument for classifying the paper as Communication and not as original research/article?

Why are these specific departments selected (i.e., general internal medicine, oncology, nephrology, rheumatology, geriatrics and pharmacy)? for some pragmatic reasons or scientifically based - more severe patients? more interested in prevention if have severe health. condition? There might be a selection bias here if offered to some severely ill patients? This should be better explained or added explanation to the limitation.

Figures 1, 2, 3 - usually when present data across time one tries to explore dynamic of an event over time (here, clicks/visits to the website) but here it remains unclear what is explored over time, i.e. what are these noticeable several picks (end November) on the graph mean - needs further explanation?

Other major issue that needs explanation is the main benefit of this approach in respect to the available material (leaflets, brochures, posters etc)? This technology to distribute (health) information seems more expensive to produce, there is necessity of internet connection (additional costs for mobile users?), particular devices (camera, internet platform, QR reader, etc). This issue needs further explanation in discussion.

Lines 210-212. “…this experiment suggests that a facilitated direct access to medical information through QR codes disseminated in health settings can help nudging people to raise their awareness on influenza and its prevention and act through a call-to-action option.” As mentioned earlier, there is no actual evaluation of the effect of this intervention (web-app usage) but rather just being (potentially) exposed to the information. Exposure doesn’t imply actually accepting and applying the knowledge toward better vaccination uptake?

Role of the funder should be better explained. Had any role in study planning, conducting, analyzing results, presenting results, decision (where) to publish, etc? This warrants further explanation.

Author Response

Please find attached our responses to your comments and a revised version of our manuscript entitled “Using ELEFIGHT® QR codes for a quick access to information on Influenza burden and prevention:  a pilot study in Lyon University Hospital”.

We were pleased to have received your pertinent comments. The modifications are in red in the revised version submitted today. Any additional requests regarding the revised version will be welcome.

All of the authors have approved the new submission of the manuscript and we assure you that it is not considered for publication elsewhere.
